Proteomic and phosphoproteomic profilings reveal distinct cellular responses during Tilapinevirus tilapiae entry and replication

Lertwanakarn Tuchakorn 1
Khemthong Matepiya 2
http://orcid.org/0000-0001-7181-7630 Setthawong Piyathip 1
Phaonakrop Narumon 3
http://orcid.org/0000-0003-3696-8390 Roytrakul Sittiruk 3
Ploypetch Sekkarin 4 sekkarin.plo@mahidol.edu
http://orcid.org/0000-0002-5707-3476 Surachetpong Win 2 5 win.s@ku.th
1 Department of Physiology, Faculty of Veterinary Medicine, Kasetsart University , Bangkok , Thailand
2 Department of Veterinary Microbiology and Immunology, Faculty of Veterinary Medicine, Kasetsart University , Bangkok , Thailand
3 Functional Proteomics Technology Laboratory, National Center for Genetic Engineering and Biotechnology , Pathum Thani , Thailand
4 Department of Clinical Sciences and Public Health, Faculty of Veterinary Science, Mahidol University , Nakhon Pathom , Thailand
5 Research Laboratory of Biotechnology, Chulabhorn Research Institute , Bangkok , Thailand
Mohanty Ipsita
Electronic publication date: 2025 Feb 21
Publication date: 2025
Volume: 13
Electronic Location ID: e18923
Received 2024 Sep 26; Accepted 2025 Jan 10
Copyright: © 2025 Lertwanakarn et al.
Copyright year: 2025
Copyright holder: Lertwanakarn et al.
License: This is an open access article distributed under the terms of the Creative Commons Attribution License, which permits unrestricted use, distribution, reproduction and adaptation in any medium and for any purpose provided that it is properly attributed. For attribution, the original author(s), title, publication source (PeerJ) and either DOI or URL of the article must be cited.
License URL: https://creativecommons.org/licenses/by/4.0/

Keywords: Tilapia lake virus, Proteomics, Phosphoproteomics, Fish cells, Viral entry

Funding: Office of the Permanent Secretary, Ministry of Higher Education, Science, Research and Innovation (OPS MHESI) Thailand Science Research and Innovation (TSRI) RGNS 65-030 Kasetsart University Research and Development Institute FF (KU) 51.67 This work was financially supported by Office of the Permanent Secretary, Ministry of Higher Education, Science, Research and Innovation (OPS MHESI), Thailand Science Research and Innovation (TSRI), Grant No. RGNS 65-030, and Kasetsart University Research and Development Institute under the project number FF (KU) 51.67. The funders had no role in study design, data collection and analysis, decision to publish, or preparation of the manuscript.

==============================
Background

Tilapia Lake virus (TiLV) poses a significant threat to global tilapia aquaculture, causing high mortality rates and severe economic losses. However, the molecular mechanisms underlying TiLV-host interactions remain largely unexplored.

Methodology

We investigated the proteomic and phosphoproteomic changes in two piscine cell lines, E-11 and RHTiB cells, following TiLV inoculation at different time points. Differential protein expressions at 10-min and 24-h post infection were selected for constructing protein-protein interactions and analyzing enriched pathways related to the viral entry and replication.

Results

Our findings revealed significant alterations in protein expression and phosphorylation states, highlighting distinct responses between the cell lines. In E-11 cells, TiLV infection suppressed proteins involved in the Janus kinase-signal transducer and activator of transcription and Fas-associated death domain protein-tumor necrosis factor receptor-associated factor pathways, leading to activation of nucleotide oligomerization domain signaling and apoptosis. In RHTiB cells, TiLV suppressed host cellular metabolism by reducing protein phosphatase activity to facilitate early viral entry, while later stages of infection showed increased activity of myosin heavy chain 9 and enhanced host immune responses via phosphorylation of ribosomal protein L17 and GTPase immunity-associated protein 7 (GIMAP7).

Conclusion

Our study suggested that TiLV employs different strategies to manipulate host cellular pathways depending on the cell type. Further studies are essential to validate these findings and ultimately facilitate the development of effective antiviral strategies.

Introduction

Tilapinevirus tilapiae, or Tilapia lake virus (TiLV), a single-stranded RNA virus belonging to the Amnoonviridae family (Koonin et al., 2023), poses a significant threat to the global tilapia aquaculture industry, as it affects 18 countries across four continents (Tran et al., 2022; Kembou-Ringert et al., 2023a). The virus contains 10 genomic segments which encode 14 functional proteins (Bacharach et al., 2016). In tilapia, infection by TiLV causes high mortality rates of up to 90% across all life stages (Surachetpong, Roy & Nicholson, 2020). Moribund fish commonly show external lesions, such as skin erosions and hemorrhage, anemia, and gill necrosis (Eyngor et al., 2014; Del-Pozo et al., 2016; Tattiyapong, Dachavichitlead & Surachetpong, 2017), as well as internal lesions, including enlargement of the hepatopancreas, spleen and kidney necrosis, gut dysbiosis, and neuroinflammation (Mojzesz et al., 2021; Turner et al., 2023; Paimeeka et al., 2024). Despite its remarkable economic impact, the molecular mechanisms underlying host–virus interactions during TiLV infection remain poorly understood. Current research on the pathogenesis of TiLV infection is primarily focused on the viral replication mechanisms, host immune responses, and subcellular processes that occur during viral infection (Lertwanakarn et al., 2021; Abu Rass et al., 2022; Lertwanakarn et al., 2023; Raksaseri et al., 2023). For example, previous studies (Mugimba et al., 2020; Piewbang et al., 2021) demonstrated the presence of viral RNA in various tissues, which indicated the multi-tissue tropisms of TiLV. Likewise, transcriptomic analysis of the livers of TiLV-infected tilapia showed potential pathways involved in viral replication and the suppression of host immune responses (Sood et al., 2021). However, in-depth studies focusing on the mechanisms underlying protein and phosphoprotein alterations in virus-infected cells remain limited.

Mass spectrometry-based proteomic and phosphoproteomic analyses have emerged as powerful tools for identifying and quantifying proteins and their phosphorylation states within infected hosts (Jean Beltran et al., 2017). Specifically, phosphoproteomic studies are valuable for understanding cellular responses to viral infections during the early stages (Yángüez et al., 2018; Hunziker & Stertz, 2022; Tang et al., 2023), as these changes occur more rapidly than those in the proteome (Li et al., 2023). Previously, both proteomic and phosphoproteomic techniques have been successfully applied to investigate host–pathogen interactions during emerging viral infections in animals and humans, such as the African swine fever virus and severe acute respiratory syndrome coronavirus 2 (SARS-CoV-2) (Keβler et al., 2018; Stukalov et al., 2021). However, the application of these techniques in the context of TiLV infection has not been explored. In fish, proteomics has provided valuable information on fish immunity (Ye, Lin & Luo, 2018) and host cellular responses to environmental stressors, such as osmotic stress (Withyachumnarnkul et al., 2024), bacterial (Causey et al., 2018; Liu et al., 2019; Zhou et al., 2020; Nissa et al., 2023) and viral infections (Xiong et al., 2011; Xu et al., 2015; Liu et al., 2020). Specifically, a proteomic study of mandarin fish brain cells infected with infectious spleen and kidney necrosis virus demonstrated alterations in glucose metabolism and the promotion of apoptosis and autophagy pathways (Wu et al., 2018). Additionally, proteomic analysis has been instrumental in identifying novel biomarkers for cardiomyopathy syndrome (Costa et al., 2021) and Aeromonas infection in Atlantic salmon (Sun et al., 2022). Based on these techniques and relevant information, in this study, we utilized proteomic and phosphoproteomic analyses to investigate the dynamics changes between the host and TiLV during the early stages of infection in two piscine cell lines, namely, E-11 and RHTiB, which has been widely used for studying TiLV entry, replication, subcellular functions, and intracellular signaling (Eyngor et al., 2014; Raksaseri et al., 2023; Lertwanakarn et al., 2023; Mohamad et al., 2024). The functional analyses yielded by the proteomic analysis offer comprehensive insights into these host–virus interactions and may contribute to a better understanding of the pathophysiology of this virus.

Materials and Methods

Portions of this text were previously published as part of a preprint (https://doi.org/10.21203/rs.3.rs-4730524/v1).

Cell culture and TiLV infection protocol

We selected two piscine cell lines, E-11 cells, derived from the snakehead fry (purchased from the European Collection Authenticated Cell Cultures), and RHTiB, brain cells derived from red hybrid tilapia (Oreochromis spp.) which has been isolated and established previously (Mohamad et al., 2024), to investigate the proteomic and phosphoproteomic profiles following TiLV infection. Both cell lines (passage 30–35) were cultured in Leibovitz’s L-15 medium, which contained 2% fetal bovine serum (Cat No. F2442; FBS, Sigma-Aldrich, Saint-Louis, MO, USA) at 25 °C without CO2. When the cells reached 80% confluency, the TiLV strain VETKU-TV08, which had been isolated from moribund fish in Pathumthani province, was diluted in L-15 without FBS and inoculated into the cells at a multiplicity of infection (MOI) of 0.1 and incubated for 10, 30 min, 1, 3, 6, and 24 h, respectively (n = 3 each). At each time point, the cytopathic effect was monitored using an inverted microscope (CKX53; Olympus, Tokyo, Japan). Cells were harvested at each time point using scrapers (Cat No. 179707PK; Thermo Fisher Scientific, Rochester, NY, USA), and 50 µL of lysates were preserved at −20 °C for viral quantification. The remainder cells were ground with an equal volume of sodium dodecyl sulfate (SDS) (Cat No. D05888; Sigma-Aldrich, Saint-Louis, MO, USA)-lysis buffer (4% w/v in 100 mM Tris/HCl, pH 8.2) containing cocktails of protease inhibitors (complete EDTA-free, Cat No. 11 873 580 001; Roche, Mannheim, Germany) and phosphatase inhibitors (Cat No. 04 906 837 001; PhosStop, Roche, Mannheim, Germany) to prevent protein degradation and dephosphorylation, respectively. The mixture was then collected and centrifuged at 10,000× g at 4 °C for 15 min. The supernatant was transferred into two new tubes for total protein and phosphoprotein analyses. Each fraction was mixed with two volumes of cold acetone and incubated overnight at −20 °C. The mixture was then centrifuged at 10,000× g for 15 min, and the supernatant was discarded. The resulting pellet was dried and stored at −80 °C until further use.

TiLV RNA quantification in cells and tissues

The E-11 and RHTiB cells were assessed for TiLV at 10 min post-infection (mpi), 30 mpi, 1 h post-infection (hpi), 3, 6, and 24 hpi using a quantitative reverse transcription polymerase chain reaction (RT-qPCR). Initially, total RNA extraction from the collected cell lysates was conducted as described by Lertwanakarn et al. (2023). Briefly, the cell lysates were mixed with GENEzol reagent (Cat No. GZR200; Geneaid Biotech, Taipei, Taiwan) and chloroform (Cat No. 288306; Sigma-Aldrich, Saint-Louis, MO, USA) and centrifuged at 15,000× g at 4 °C. The supernatant was collected and treated with DNase I (Cat No. AM2224; Thermo Fisher Scientific, Carlsbad, CA, USA), followed by 2-propanol (Cat No. 109634; Merck, Darmstadt, Germany), and stored at −20 °C for 2 h. After thawing and centrifugation, the pellets were washed with ethanol and air-dried. The collected RNA was then reconstituted with RNase-free water and converted to complementary deoxyribonucleic acid (cDNA) using the ReverTra Ace cDNA synthesis kit (Cat No. FSQ-201; Toyobo, Osaka, Japan). Briefly, one microgram of the RNA was added into 20 μL reaction mixture containing 2 μM Oligo d(T), 0.5 mM dNTPs mix and 100 U of reverse transcriptase, and the reactions were incubated as follow: 65 °C for 5 min, 42 °C for 60 min and 85 °C for 5 min in the T100 thermal cycler (Bio-Rad, Foster city, CA, USA). Finally, the TiLV viral copy number was assessed using a SYBR Green-based qPCR assay. The assay was carried out in a 20 μL reaction mixture containing 10 μL of iTaq universal SYBR green supermix (Cat No. 172–5125; Bio-Rad, Hercules, CA, USA), 0.3 μL of forward (CTGAGCTAAAGAGGCAATATGGATT) and reverse (CGTGCGTACTCGTTCAGTATAAGTTCT) primers, 4 μL of cDNA template and molecular-grade water to adjust the final volume. The cycling condition was set as follows: denaturation at 95 °C for 3 min, 40 cycles of 95 °C for 10 s, and 60 °C for 30 s (Tattiyapong, Sirikanchana & Surachetpong, 2018). At the end of the qPCR cycle, the TiLV log copy number (PCR product size 112 bp) was retrieved from the standard curve of the melting temperature obtained using CFX Maestro Software (Bio-Rad, Chicago, IL, USA).

Detection of TiLV in cell lines using an immunofluorescence assay

In line with the protocol described by Mohamad et al. (2024), an immunofluorescent (IFA) assay was performed to study the dynamics of TiLV infection in the RHTiB cells. Briefly, 1 × 105 RHTiB cells were seeded on a cell culture chamber slide (SPL Life Sciences, Gyeonggi-do, Korea) and cultured in an L-15 medium supplemented with 5% FBS at 25 °C until they reached 80–90% confluency. Subsequently, the cells were washed and inoculated with a control medium or 0.1 MOI of TiLV for 10, 30 min, 1, 3, or 24 h at 25 °C. Following incubation, the samples were fixed with ice-cold 100% methanol for 10 min, followed by permeabilization with 0.1% Triton X-100 (Cat No. T8532; Sigma-Aldrich, Saint-Louis, MO, USA) for 10 min. A blocking solution containing 2% bovine serum albumin (Cat No. 85040C; Sigma-Aldrich, Darmstadt, Germany) in phosphate buffer saline (PBS) was applied for 30 min to reduce non-specific binding. Subsequently, the cells were probed with a 1:100 dilution of rabbit polyclonal anti-TiLV IgG prepared in blocking solution overnight at 4 °C. The cells were then washed three times with PBS and incubated with a goat anti-rabbit IgG-Alexa Fluor 488 (Cat No. ab190195; Abcam, Carlsbad, CA, USA) diluted 1:1,000 in PBS for 1 h at room temperature. Finally, the cell nuclei were stained with 1 µg/mL of diaminophenylindole (Cat No. 10 236 276 001; DAPI, Sigma-Aldrich, St Louis, MO, USA) before visualization under a confocal microscope (Fluoview 3000; Olympus, Tokyo, Japan). The negative control cells probed with secondary IgG was conducted in parallel to confirm that non-specific signal had occurred in the experiment (Fig. S1).

Mass spectrometry-based proteomic and phosphoproteomic analyses

Phosphoprotein enrichment and sample preparation

The phosphoproteins in the samples were enriched using the immobilized metal affinity column resin charged with Ga3+ ions as per the manufacturer’s instructions (Cat No. 90003; Pierce, Thermo Fisher Scientific, Rockford, IL, USA). Each sample was adjusted to a final concentration of 0.5 mg/mL using lysis/binding/wash buffer. The column was pre-equilibrated with lysis/binding/wash buffer containing 3-[(3-cholamidopropyl) dimethylammonio]-1-propanesulfonate (CHAPS; Cat No. 436550250; Thermo Fisher Scientific, Rockford, IL, USA) and centrifuged at 1,000× g for 1 min at 4 °C. The samples were applied to the column and incubated for 30 min at 4 °C. Following incubation, the column was centrifuged at 1,000× g for 1 min at 4 °C to collect the flow-through fraction. After three washes with lysis/binding/wash buffer containing CHAPS, the phosphoproteins were eluted from the column via incubation with 1 mL of elution buffer at room temperature for 3 min, then centrifuged at 1,000× g for 1 min at 4 °C. The elution process was repeated four times, and the collected fractions were pooled for liquid chromatography and mass spectrometry (LC-MS/MS) analysis.

The protein concentrations in the samples were assessed using the modified Lowry technique (Waterborg & Matthews, 1994). All the protein and phosphoprotein samples were prepared by adding 5 mM dithiothreitol (Cat No. 646563; Sigma-Aldrich, Saint Louis, MO USA) to 10 mM ammonium bicarbonate (NH4HCO3) (Cat No. A6141; Sigma-Aldrich, Saint Louis, MO, USA) at 60 °C for 1 h to reduce the disulfide-containing compounds. Alkylation of the reduced cysteine residues was achieved by incubating with 15 mM iodoacetamide (Cat No. I6125; Sigma-Aldrich, Saint Louis, MO, USA) in 10 mM ammonium bicarbonate at 25 °C for 45 min. Subsequently, the proteins and phosphoproteins were digested with trypsin (Cat No. V5111; Promega, Madison, WI, USA) for 3 h at room temperature. Finally, the digested peptide samples were dissolved in 0.1% formic acid (FA) (Cat No. 270480100; Thermo Fisher Scientific, Waltham, MA, USA) and submitted for LC-MS/MS analysis.

LC-MS/MS analysis

The identification of the digested peptides was carried out using the Ultimate 3000 Nano/Capillary LC System (Thermo Fisher Scientific, Waltham, MA, USA) coupled to a ZenoTOF 7600 mass spectrometer (SCIEX, Framingham, MA, USA). The digested peptide samples were concentrated using Acclaim 5 µm PepMap 300 µ-Precolumns packed with C18 (300 µm × 5 mm, 5 µm, 100 Å, Thermo Fisher Scientific, Waltham, MA, USA) and subsequently separated using an Acclaim PepMap Rapid Separation Liquid Chromatography column (75 μm × 15 cm, 2 µm, 100 Å, Thermo Fisher Scientific, Waltham, MA, USA). The C18 column was enclosed within a temperature-controlled column oven set to 35 °C. For protein separation, the solvents A and B, which contained 0.1% FA in water and 0.1% FA in 80% acetonitrile, respectively, were utilized. A gradient of solvent B ranging from 5% to 55% over 30 min was employed at a flow rate of 0.30 μL/min. The source and gas parameters on the ZenoTOF 7600 system were configured as follows: ion source gas 1 at eight pounds per square inch (psi), curtain gas at 35 psi, CAD gas at 7 psi, source temperature at 200 °C, polarity set to positive, and spray voltage at 3,300 V. For data-dependent acquisition (DDA), the top 50 most abundant precursor ions per survey MS1 were selected for subsequent MS/MS analysis. These precursor ions met an intensity threshold exceeding 150 counts per second (cps). The precursor ions were dynamically excluded for 12 s after two incidences of MS/MS sampling (with dynamic collision energy enabled). The MS2 spectra were collected in the mass range of 100–1,800 m/z with a 50-ms accumulation time. The collision energy (CE) parameters consisted of an 80 V declustering potential (DP), no DP spread, and a CE spread of 0 V. The time bins, which incorporated all the channels, were summed using a Zeno trap threshold of 150,000 cps. The cycle time for the top 60 DDA method was 3.0 s. To ensure quality control throughout the analytical process, three replicates of the same sample were analyzed to monitor the reproducibility of the result. Additionally, the digestion of bovine serum albumin served as a quality control sample to assess the performance and reliability of the mass spectrometry instrument and the entire analytical workflow, as previously reported (Bittremieux et al., 2018; Vincent et al., 2019).

Data processing and analysis

For protein identification, the raw mass spectral data were processed using MaxQuant software (version 2.2.0.0) for peptide and protein identification. MS/MS searches were performed against the reference proteome database for Oreochromis niloticus, NCBI:txid8128 (downloaded from UniProt), which includes 76,021 entries, supplemented with a contaminant database. The significant threshold for protein identification was established with a p-value less than 0.05, and a false discovery rate (FDR) was set to 1% for both the peptides and proteins. The minimum peptide length was set to seven amino acid residues containing at least one unique peptide, as outlined in previous studies (Cottingham, 2009; Vincent et al., 2019). Label-free quantification was performed using the MaxLFQ algorithm integrated into MaxQuant with a minimum ratio count of 2. For the phosphoproteomic analysis, phosphorylation of the serine, threonine, and tyrosine residues were included as variable modifications. The protein and phosphorylation site localizations were assessed using the MaxQuant software-supported post-translational modification localization probability algorithm.

Bioinformatics analysis

The protein and phosphoprotein abundance in the E-11 and RHTiB cells were compared across all time points using Metaboanalyst.ca (version 6.0), a web-based platform for statistical and pathway analysis of omics data. Analysis of variance (ANOVA) was utilized to identify statistically significant differences of proteins and phosphoproteins between the groups and among different time point of infection, while controlling FDRs and ensuring a significance level of <0.05 (Gupta & Pevzner, 2009). Gene Ontology enrichment analysis was conducted to reveal the upregulation of the biological processes, molecular functions, and cellular components of the differentially expressed proteins and phosphoproteins. Cluster analyses and heatmap visualizations were employed through Metaboanalyst.ca to enhance the understanding of the protein expression and phosphorylation dynamics throughout the TiLV infection timeline. Protein–protein interaction (PPI) networks were constructed from the differential expressed proteins at each time point using the Search Tool for Interactions of Chemicals (STITCH database; http://stitch.embl.de) (Szklarczyk et al., 2016) to identify the cellular mechanisms involved in TiLV infection. Pathway analysis was performed using the Kyoto Encyclopedia of Genes and Genomes (KEGG) database (Kanehisa & Goto, 2000), with confidence levels ranging from low edge (>0.15) to highest edge confidence scores (>0.900) to represent the key signal transduction pathways and metabolic processes impacted by TiLV infection in the host cells. The pathway analysis was validated using ShinyGO 0.80 tool (bioinformatics.sdstate.edu/go/) to confirm the key pathways altered during TiLV infection (Ge, Jung & Yao, 2020).

Results

Dynamics of TiLV in the E-11 and RHTiB cell lines

We investigated the dynamics of TiLV in the E-11 and RHTiB cell lines using morphological observations, qRT-PCR for viral RNA quantification, and IFA (Fig. 1). Up to 24 h post TiLV infection, neither cell line had exhibited cytopathic effects or morphological changes (Fig. 1A). Initially, TiLV RNA was detected at 10 mpi in both cell lines, with similar viral copy numbers of 6.71 ± 0.03 and 6.25 ± 0.18 log10 copies/400 ng cDNA, respectively (Fig. 1B). Subsequently, the viral load increased gradually in both cell lines, peaking at 24 hpi with viral copy numbers of 8.22 ± 0.06 and 6.76 ± 0.04 log10 copies/400 ng cDNA in the E-11 and RHTiB cells, respectively. Notably, the viral concentration in the E-11 cells was significantly higher than that in the RHTiB cell lines at 6 and 24 hpi (p < 0.05). Additionally, TiLV was detected in the RHTiB cells throughout the studied period using IFA. Interestingly, an intracellular signal representing TiLV infection was initially observed at 10 mpi, while colocalization of TiLV with cell nuclei was first detected at 30 mpi (Fig. 1C).

Figure 1 Dynamics of TiLV infection in the E-11 and RHTiB cell lines.

(A) Microscopic examination of the control and tilapia lake virus (TiLV)-infected E-11 and RHTiB cells at 24 h post-infection (hpi). No cytopathic effects or morphological changes were observed in either cell line. (B) TiLV RNA concentrations quantified from the infected E-11 and RHTiB cell lines at various time points. *p < 0.05 compared to 10 min post-infection (mpi); Δ p < 0.05 compared to 30 mpi; ψ p < 0.05 compared to 1 hpi; φ p < 0.05 compared to 3 hpi; ξ p < 0.05 compared to 6 hpi; ℘ p < 0.05 compared to the E-11 cells at the same time point. (C) Representative confocal micrographs of the infected RHTiB cell lines probed with specific antibodies against TiLV conjugated with Alexa Fluor 488 staining (green color) at different time points. The nuclei were stained with diaminophenylindole (DAPI, blue color). Positive TiLV signals (red arrow) were detected throughout the study period.

Proteomic and phosphoproteomic analyses of the E-11 and RHTiB cells

The proteomic and phosphoproteomic patterns of the E-11 and RHTiB cells were analyzed using in-solution digestion and LC-MS/MS (Fig. 2). Notably, we identified remarkable changes (2,057 proteins and 1,838 phosphoproteins) in the E-11 cells, while a total of 2,473 proteins and 2,157 phosphoproteins were identified in the RHTiB cells (Tables S1, S2). Partial least squares discriminant analysis (PLS-DA) of the proteomic data revealed distinct protein profiles between the E-11 and RHTiB cells (Fig. 2A). In contrast, cluster analyses using PLS-DA of the phosphoproteins showed an overlap between the E-11 and RHTiB cells, which suggested some similar phosphorylation patterns between the two cell types (Fig. 2B). Additionally, the heat maps display the dynamic expression levels of protein (Fig. 2C) and phosphoprotein (Fig. 2D) in the E-11 and RHTiB cell lines throughout the study. Along the vertical axis, the hierarchical clustered by the dendrogram highlighted the distinct patterns of protein and phosphoprotein expression between the two cell lines following TiLV infection. Meanwhile, the horizontal axis illustrated the differential changes of protein and phosphoprotein patterns at each time point.

Figure 2 Proteomic and phosphoproteomic analyses comparing E-11 and RHTiB cells.

(A) Partial least square discrimination analysis (PLS-DA) of the proteomic data reveals distinct clusters for both cell lines. (B) PLS-DA of the phosphoproteomic profiles displaying overlaps between the two clusters. (C) Heatmap analysis comparing the proteins of the E-11 and RHTiB cells. (D) Heatmap analysis comparing the phosphoprotein profiles of the E-11 and RHTiB cells.

ANOVA and enrichment pathway analyses of the TiLV-infected E-11 cells

The dynamics of the protein and phosphoprotein alterations in the TiLV-infected E-11 cells were analyzed using ANOVA and PLS-DA to identify differentially expressed proteins and phosphoproteins (Fig. 3). The analysis revealed significant changes in 53 proteins and 136 phosphoproteins throughout the study period (Figs. 3A, 3B, Tables S3, S4). Notably, the control E-11 cells formed a separate cluster, which demonstrated a distinct protein profile compared to the infected cells (Fig. 3C). Conversely, PLS-DA analysis of the phosphoproteome data did not reveal distinct clusters among the E-11 cells collected at different time points (Fig. 3D).

Figure 3 Bioinformatic analyses of TiLV-infected E-11 cells across all time points.

Analysis of variance (ANOVA) test demonstrating significant differential expression of (A) 53 proteins and (B) 136 phosphoproteins. Clustering analyses using a partial least square discrimination analysis (PLS-DA) map illustrating distinct patterns of (C) proteins expression and (D) phosphoproteins in the E-11 cells.

To explore the viral entry and replication mechanisms as well as the cellular response processes, PPI networks were constructed from differentially altered significant proteins and phosphoproteins at 10 and 24 mpi using the STITCH online software, and the enrichment pathways were analyzed against the KEGG database. At 10 mpi, when TiLV signals were initially detected in the E-11 cells, 24 proteins and 33 phosphoproteins were significantly decreased (Table S5). The pathway analysis from the PPI of these proteins demonstrated relationships between basic helix-loop-helix family, member e41 (BHLHE41), calcium-binding and coiled-coil domain-containing protein 2 (CALCOCO2), transgelin (TAGLN), eukaryotic translation elongation factor 2 (EEF2), exosome component 3 (EXOSC3), prolyl 4-hydroxylase (P4HB), ADAM metallopeptidase domain 12 (ADAM12), and steroidogenic factor 1 (SF1), as well as the phosphoproteins NMT1, receptor-interacting protein kinase 1 (RIPK1), exportin-5 (XPO5), class II transactivator (CIITA), plakophilin-9 (PKP9), protein tyrosine phosphatase receptor type C (PTPRC), endonuclease G (ENDOG), Janus kinase-2 (JAK2), Ras-related nuclear protein (RAN), myosin phosphatase Rho interacting protein (MPRIP), Rho GTPase activating protein 21 (ARHGAP21), Rho guanine nucleotide exchange factor 12 (ARHGEF12), and caspase 1 (CASP1) (Fig. 4A). Specifically, the KEGG pathway analyses showed immediate suppression of several intracellular mechanisms, including RNA degradation, the JAK–signal transducer and activators of transcription (STAT) signaling pathway, and Fas-associated death domain protein (FADD)–tumor necrosis factor receptor associated factor (TRAF) (Table S6). At 24 hpi, one protein and 51 phosphoproteins gradually increased in the E-11 cells (Table S5). The constructed PPI network revealed relationships among the phosphoproteins, such as enolase 1 (ENO1), P4HB, muscle Ras oncogene (MRAS), Lamin-B2 (LMNB2), Ecto-NOX disulfide-thiol exchanger 1 (ENOX1), telomerase reverse transcriptase (TERT), thioredoxin interacting protein (TXNIP), tumor necrosis factor receptor-associated factor 2 (TRAF2) hydroxysteroid dehydrogenase like 1 (HSDL1), NMD3 ribosome export adaptor (NMD3), Epstein-Barr nuclear antigen 1 binding protein 2 (EBNA1BP2), SERTA domain-containing protein 2 (SERTAD2), caspase 9 (CASP9), intraflagellar transport 122 (IFT122), lysophosphatidylglycerol acyltransferase 1 (LPGAT1), golgin subfamily A member 4 (GOLGA4), histone proteins (H3F3B, HIST1H2BD), cluster of differentiation 97 (CD97), and calbindin 2 (CALB2) (Fig. 4B). These proteins participated in nucleotide oligomerization domain (NOD)-like receptor signaling and apoptotic pathways in the fish cells (Table S6).

Figure 4 Protein–protein interaction network in Tilapia lake virus (TiLV)-infected E-11 cells.

The diagrams were constructed from differentially expressed proteins and phosphoproteins at (A) 10 mpi and (B) 24 hpi using STICH software version 5.0 (http://stitch.embl.de/). The pathway analyses were conducted using KEGG database and ShinyGo 0.80 software and highlighted within the diagram.

ANOVA and enrichment pathway analyses of the TiLV-infected RHTiB cells

The proteins and phosphoproteins obtained from the TiLV-infected RHTiB cells at each time point were analyzed using ANOVA and clustering methods (Fig. 5). The ANOVA analysis identified differential expressions of 71 proteins and five phosphoproteins across the study (Figs. 5A, 5B, Tables S7, S8). The PLS-DA plot of the proteomic data successfully separated the TiLV-infected RHTiB cells at each time point into distinct clusters (Fig. 5C). Similarly, the control RHTiB cells formed an isolated cluster, which resembled the control E-11 cells, and represented a unique protein profile compared to the infected cells. Conversely, the PLS-DA analysis of the phosphoproteomic profile did not reveal distinct clusters among the RHTiB groups at any time point (Fig. 5D).

Figure 5 Bioinformatic analyses of TiLV-infected RHTiB cells at different time points.

Analysis of variance showing differential expressions of (A) 71 proteins and (B) 11 phosphoproteins. Partial least square discrimination analysis (PLS-DA) clustering methods from (C) differential proteins and (D) phosphoproteins expression in the RHTiB cells.

Based on the differential expression of the proteins and phosphoproteins in the RHTiB cells, we focused on the 10 mpi and 24 hpi groups to evaluate the possible mechanisms involved in the TiLV entry, replication, and host cellular responses. At 10 mpi, when the TiLV signals were first detected in the RHTiB cells, four proteins, namely, phosphatase 2 regulatory subunit B alpha (PPP2R2A), ATP-binding cassette sub-family C member 5 (ABCC5), methylenetetrahydrofolate dehydrogenase 2 (MTHFD2), and Tetraspanin-13 (TSPAN13), and one phosphoprotein, BCAS3, were significantly decreased compared to the control group (Table S9). The enrichment analysis from these molecules suggested the suppression of adenosine monophosphate-activated protein kinase (AMPK) signaling in the RHTiB cells at this time point (Fig. 6A, Table S10). At 24 hpi, we observed alterations in four proteins, that is, myosin heavy chain 9 (MYH9), pleckstrin homology domain-containing protein 2 (PLEKHH2), Tensin 3 (TNS3), and methylenetetrahydrofolate dehydrogenase 1 (MTHFD1), and two phosphoproteins, GTPase immunity-associated protein family member 7 (GIMAP7) and phosphorylated ribosomal protein L17 (RPL17). The relationships between these proteins indicated increased activity of the actin cytoskeleton in the TiLV-infected RHTiB cells (Fig 6B, Table S10).

Figure 6 Protein–protein interaction network in TiLV-infected RHTiB cells.

The diagrams were constructed from differentially expressed proteins and phosphoproteins at (A) 10 mpi and (B) 24 hpi using STICH software version 5.0 (http://stitch.embl.de/). The pathway analyses were conducted using KEGG database and ShinyGo 0.80 software and highlighted within the diagram.

Discussion

Viruses are microorganisms that require host cells for their replication (Villanueva, Rouillé & Dubuisson, 2005). Hence, studying virus–host cell interactions allow the impacts of viruses on host cellular regulation to be understood and provide insights into how host cells combat viral infections (Hoenen & Groseth, 2022). Generally, host cells respond to viral infections through a variety of mechanisms, such as immune activation, metabolic alteration, and cell cycle arrest, which demonstrates the dynamic nature of virus–host interactions (Katze, He & Gale, 2002; Bagga & Bouchard, 2014; Raque et al., 2023). TiLV, a novel RNA virus identified in tilapia, can infect multiple fish cell lines and induce cell death across a range of tissues, including tilapia brain-, heart-, and liver-derived cell lines (Tattiyapong, Sirikanchana & Surachetpong, 2018; Thangaraj et al., 2018; Nanthini et al., 2019; Yadav et al., 2021; Li et al., 2022) and other piscine cell lines (Tattiyapong, Dachavichitlead & Surachetpong, 2017; Raksaseri et al., 2023; Li et al., 2022). Despite advances in identifying the broad tropisms of the virus, the mechanisms of TiLV entry, replication, and specific host cell responses pose significant research challenges. In this study, we applied proteomic and phosphoproteomic analyses to understand early host–virus interactions during TiLV infection in two cell lines, E-11 cells, derived from the snakehead fish, which are not the natural host of TiLV but has been extensively studied in previous research (Eyngor et al., 2014; Tattiyapong, Dachavichitlead & Surachetpong, 2017; Lertwanakarn et al., 2021), and RHTiB cells, the primary brain cells from the red tilapia that can propagate the virus (Mohamad et al., 2024). Our findings using RT-qPCR, and an IFA assay confirmed the early detection of TiLV in RHTiB cells at 10 mpi. Moreover, the presence of TiLV in the cell nuclei at 30 mpi indicated successful viral entry and the utilization of the host machinery for replication. These observations are consistent with those of previous studies, which showed early TiLV detection in fish cells within 1 hpi using different methods (Abu Rass et al., 2022; Piewbang et al., 2022; Lertwanakarn et al., 2023). Similarly, our results support previous findings indicated that the virus can enter fish cells rapidly—within a few minutes—and potentially cause biological changes in the host cells. Additionally, these findings support the selected time frame for investigating cellular damage at multiple levels in infected cells. Interestingly, in addition to the observed cellular damage, our findings revealed variations in the infection dynamics and TiLV replication across different fish cells. Notably, the TiLV load in the E-11 cells was higher than that in the RHTiB cells at 24 hpi, which is consistent with the results of a previous study (Lertwanakarn et al., 2023). We propose that while TiLV is capable of propagating in recently established RHTiB cells, these cells are less susceptible to TiLV compared to well-established E-11 cells. This reduced susceptibility could be attributed to variations in the availability of receptors or the cellular machinery necessary for TiLV replication (Li et al., 2022; Lertwanakarn et al., 2023; Mohamad et al., 2024). To support this hypothesis, the bioinformatic analyses presented in this study demonstrated distinct patterns in the protein and phosphoprotein responses between the E-11 and RHTiB cells, which suggests differences in the host response mechanisms. Comprehensive analysis of the important signaling pathways in both cell lines is essential to further elucidate the mechanisms of viral replication and host response.

Bioinformatic analyses of the infected E-11 cells revealed unique shifts in the patterns of protein expression during TiLV infection. At 10 mpi, there was a significant suppression of the proteins associated with the JAK family and STAT pathways, such as Janus kinase 2 (JAK2), protein tyrosine phosphatase (PTPRC), and the major histocompatibility complex CIITA. Proteins associated with the FADD–TRAF pathways, including CASP1 and RIPK1, were also suppressed. Importantly, these pathways, particularly JAK–STAT and FADD–TRAF, are critical for cellular responses to inflammation and interferon production during viral infections (Hsu et al., 1996; Liongue et al., 2012; Ezeonwumelu, Garcia-Vidal & Ballana, 2021). In fish, these pathways are known to facilitate interferon (IFN)-γ production through the activation of tyrosine phosphatases and mitogen-activated protein kinase (MAPK) signaling pathways, which lead to the release of pro-inflammatory cytokines and antiviral proteins (Sobhkhez et al., 2013; Lee et al., 2022). Therefore, the observed downregulation of the proteins in these pathways indicates that TiLV may interfere with interferon-mediated antiviral activity to facilitate viral entry and replication, as seen with other fish viruses (Xu, Evensen & Munang’andu, 2016; Guo et al., 2022). The suppression of RIPK1 suggests that TiLV also prevents nuclear factor kappa B pathway (NF-κB) activation and subsequently reduces MAPK signaling and IFN production and promotes cell death (Udawatte & Rothman, 2021). These findings are supported by the increased levels of phosphoproteins involved in the apoptotic pathway (SERTAD, TXNIP, CASP9), NOD-like receptor signaling (TXNIP, TRAF2), and cell replication cycle (H3F3B, HIST1H2BD, TERT) observed in the TiLV-infected E-11 cells in our study at 24 hpi. Similarly, transcriptomic profiling of liver tissues from TiLV-infected fish revealed that TiLV may be recognized through the NOD-like receptor, leading to the upregulation of traf2 and the activation of the NF-κB pathway (Sood et al., 2021). Interestingly, other piscine viruses utilize histone proteins to manipulate host cellular functions and suppress antiviral mechanisms, thereby enhancing viral replication (Tarakhovsky & Prinjha, 2018). Additionally, these viruses interact with TERT and telomeric functions, which induces cellular stress that leads to senescence and apoptosis in the infected cells (Salimi-Jeda et al., 2021). Although TiLV-induced apoptosis has not been well documented, a study demonstrated that infected E-11 cells displayed cytopathic effects (CPEs) together with mitochondrial damage and ATP depletion, which suggests similar interference with cellular integrity (Raksaseri et al., 2023). Furthermore, while no CPEs were detected in E-11 cells after 24 hpi in this study, alterations in proteins and phosphoproteins involving in apoptotic pathway aligned with other reports which have highlighted remarkable CPEs in E-11 cells following 2 days post infection (Lertwanakarn et al., 2021; Raksaseri et al., 2023). Based on these findings, we hypothesize that the mechanism underlying TiLV infection in E-11 cells involves the suppression of the JAK–STAT and FADD–TRAF pathways. This suppression not only promotes viral replication but may facilitate the apoptotic process (Fig. 7A). Hence, targeting proteins associated with the JAK–STAT and cell replication pathways is crucial in controlling TiLV replication during the early stages of infection. Given the complex interplay of these pathways in viral pathogenesis, further studies on antiviral therapies are warranted. These studies should aim to explore and address the specific viral mechanisms that induce cell death, as has been previously outlined (Lertwanakarn et al., 2021; Zhang et al., 2021). Such research could provide significant insights into potential therapeutic interventions for TiLV and other similar viruses.

Figure 7 A graphical summary illustrating the dynamics of Tilapia lake virus (TiLV) infection and potential cellular responses in different cell lines.

(A) In E-11 cells, TiLV suppresses FADD–TRAF and JAK–STAT pathway at 10 mpi, significantly inhibiting RIPK, and subsequently activates NOD-like receptor signaling and apoptotic pathway at 24 hpi. (B) In RHTiB cells, TiLV suppresses the AMPK pathway 10 mpi, leading to viral replication and the promotion of cytoskeletal activities and antiviral response at 24 hpi.

In the RHTiB cells, infection by TiLV prompted a distinct cellular response compared to the E-11 cells. Specifically, at 10 mpi, there was a suppression of the protein PPP2R2A, an enzyme involved in the dephosphorylation of intracellular proteins (Gerlt et al., 2021), and MTHFD2, which plays a role in RNA metabolism and translation (Koufaris & Nilsson, 2018). Previous research has demonstrated that the downregulation of protein phosphatase leads to cell senescence in aging zebrafish neuronal cells (Xing et al., 2023), and the suppression of MTHFD2 has been linked to cell death in infectious hematopoietic necrosis virus (IHNV)-infected zebrafish larvae (Briolat et al., 2014). Thus, the suppression of these proteins in infected RHTiB cells indicates that TiLV may inhibit host cell replication and metabolism early in the infection process while maintaining cellular protein phosphorylation, which facilitates viral entry and replication, as seen with other viruses, such as influenza A virus, hepatitis B, and Epstein–Barr virus (EBV) (Pim et al., 2005; Garibal et al., 2007; Gerlt et al., 2021; Xi, Luckenbaugh & Hu, 2021). Additionally, at 10 mpi, the TiLV-infected RHTiB cells showed inhibition of cytoskeletal proteins, including BCAS3, which controls the direction of cell migration (Qiu et al., 2018). Similar suppression of cytoskeletal proteins associated with the RhoA GTPase pathway, including plakophilin-3 (PKP3), ARHGEF12, ARHGAP21, MPRIP, and TAGLN, has been observed in infected E-11 cells. Cytoskeletal proteins, particularly microtubules, are targeted by viruses to facilitate their entry mechanisms (Ploubidou & Way, 2001; Liu et al., 2018; Walsh & Naghavi, 2019; Liqun et al., 2020), such as endosomal formation (Liu et al., 2024b) and micropinocytosis (Levicán-Asenjo et al., 2019). For instance, TiLV enters primary cells derived from the bulbus arteriosus of O. mossambicus through dynamin activity but not endosomal acidification (Abu Rass et al., 2022). However, the downregulation of these cytoskeletal proteins in both E-11 and RHTiB cells suggests that the early TiLV entry mechanism may rely on other processes, such as receptor-mediated or transmembrane diffusion (Kembou-Ringert et al., 2023b) (Fig. 7B).

In contrast, after 24 hpi, the infected RHTiB cells showed increased activity of cytoskeletal proteins such as MYH9 and PLEKHH2 (Matozo, Kogachi & de Alencar, 2022). MYH9 has been identified as a receptor for the herpes simplex virus 1 and EBV (Liu et al., 2024a) and serves as a co-receptor in the entry process of SARS-CoV-2 (Clausen et al., 2020). Additionally, MYH9 is essential for the cellular entry of infectious pancreatic necrosis virus (Shao, Zhao & Tang, 2021) and the replication of porcine reproductive and respiratory syndrome virus (Gao et al., 2016). Hence, the mechanism of TiLV proliferation in RHTiB cells may involve the cytoskeletal activity to facilitate viral entry into these susceptible cells. The discrepancy results between the cytoskeletal proteins at 10 mpi and 24 hpi in RHTiB-infected cells demonstrates that at different time course of infection, the virus may utilize different pathways and in depth-analysis should be warranted to fulfil the viral entry and replication mechanisms. In our study, we observed an increase in RPL17 and GIMAP7 in the infected RHTiB cells at 24 hpi. The phosphorylation of these proteins is associated with the concentration of infectious salmon anemia virus in the head kidney of infected salmon (Gervais et al., 2022), whereas the phosphorylation of GIMAPs plays a crucial role in the antiviral response in zebrafish (Balla et al., 2020). These findings suggest that TiLV modulates the host immune response and promotes significant viral replication in RHTiB cells (Fig. 7B), which is consistent with the results of previous studies (Sobhkhez et al., 2013; Mugimba et al., 2020; Lertwanakarn et al., 2023). Further investigations into immunomodulatory agents may therefore help reduce viral replication and help combat this widespread disease in tilapia.

One of the primary limitations of our study was the lack of validation for protein expression, which could have further strengthened the reliability of this study. Indeed, many commercial antibodies are developed for mammalian species, often resulting in lower specificity and limited cross-reactivity with tilapia proteins (Dixon, Barreda & Sunyer, 2018). While developing antibodies specifically for tilapia proteins could be considered as an alternative approach, this process is resource-intensive and faces challenges related to sensitivity and consistency as described in comprehensive review (Dixon, Barreda & Sunyer, 2018). Other molecular techniques, such as targeting pro- and antiviral genes with siRNA and CRISPR/Cas9, could serve as complementary methods for validating proteomic results. Nonetheless, the use of specific antibodies may still be necessary for definitive confirmation of gene manipulation effects (Boyer et al., 2013; Dolgalev & Poverennaya, 2021). To ensure the accuracy of our protein identifications, we conducted bioinformatics analyses using multiple tools, including STITCH and ShinyGO 0.80, which yielded consistent results. Furthermore, the biological and functional roles of these proteins, particularly those involved in apoptosis, were reported in the existing literature (Raksaseri et al., 2023). Further in-depth studies using advanced tools such as kinase-substrate enrichment analysis (KSEA) or other omics approaches, including transcriptomics, should be employed to map the regulatory kinases involved and to elucidate broader signaling networks. These iterative approaches are expected to generate a more detailed timeline of virus-induced phosphorylation events and to identify potential host targets for therapeutic interventions.

Conclusions

Understanding the early interactions and subsequent cellular changes between viruses and host cells is important for advancing the knowledge of viral pathogenesis. In this article, we have provided valuable insights into the distinct proteomic and phosphoproteomic alterations between two piscine cell lines, E-11 and RHTiB, during TiLV infection. Our results show that TiLV-infected E-11 cells exhibit significant changes in the proteins involved in the JAK–STAT and FADD–TRAF pathways during early infection, which further activate NOD-signaling and apoptotic pathways, leading to the viral replication and cell death. In contrast, in RHTiB cells, TiLV infection suppresses host cellular metabolism to facilitate viral entry during early infection, while later stages of viral replication require cytoskeletal proteins and promote host immune responses. Further experimental studies are essential to validate these hypotheses and elucidate the precise mechanisms by which these proteins influence TiLV infection in different cell lines. Additionally, integrating omics approaches, such as transcriptomics, proteomics, and functional assays, can provide a comprehensive understanding of the molecular interactions underlying TiLV pathogenesis and host–virus interactions and ultimately facilitate the development of effective antiviral strategies.

Supplemental Information

Supplemental Information 1 Raw data of mass-spectrometry-based proteomic analysis of E-11 and RhTiB cells at different time points.

Supplemental Information 2 Raw data of mass-spectrometry-based phosphoproteomic analysis of E-11 and RhTiB cells at different time points.

Supplemental Information 3 Analysis of variance comparing differential protein expression in TiLV-infected E-11 cells across the study.

Supplemental Information 4 Analysis of variance comparing differential phosphoprotein expression in TiLV-infected E-11 cells across the study.

Supplemental Information 5 Altered protein and phosphoproteins in TiLV-infected E-11 cells at 10 mpi and 24 hpi.

Supplemental Information 6 Pathway analysis from selected altered proteins and phosphoproteins in TiLV-infected E-11 cells.

Supplemental Information 7 Analysis of variance comparing differential protein expression in TiLV-infected RHTiB cells across the study.

Supplemental Information 8 Analysis of variance comparing differential phosphoprotein expression in TiLV-infected RHTiB cells across the study.

Supplemental Information 9 Altered protein and phosphoproteins in TiLV-infected RHTiB cells at 10 mpi and 24 hpi.

Supplemental Information 10 Pathway analysis from selected altered proteins and phosphoproteins in TiLV-infected RHTiB cells.

Supplemental Information 11 Immunofluorescence micrograph of RHTiB cell stained with only goat anti-rabbit IgG-Alexa Fluor 488.

The nuclei were counterstained with DAPI.

We would like to specially thank all personnels in Functional Proteomics Technology Laboratory, National Center for Genetic Engineering and Biotechnology, and Faculty of Veterinary Science for the facility and laboratory support.

Additional Information and Declarations

Competing Interests

The authors declare that they have no competing interests.

Author Contributions

Tuchakorn Lertwanakarn conceived and designed the experiments, analyzed the data, prepared figures and/or tables, authored or reviewed drafts of the article, and approved the final draft.

Matepiya Khemthong performed the experiments, prepared figures and/or tables, and approved the final draft.

Piyathip Setthawong analyzed the data, authored or reviewed drafts of the article, and approved the final draft.

Narumon Phaonakrop performed the experiments, prepared figures and/or tables, and approved the final draft.

Sittiruk Roytrakul conceived and designed the experiments, analyzed the data, prepared figures and/or tables, and approved the final draft.

Sekkarin Ploypetch conceived and designed the experiments, analyzed the data, prepared figures and/or tables, authored or reviewed drafts of the article, and approved the final draft.

Win Surachetpong conceived and designed the experiments, authored or reviewed drafts of the article, and approved the final draft.

Data Availability

The following information was supplied regarding data availability:

The raw data are available in the Supplemental Tables.

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
