# Peer review of "Proteomic and phosphoproteomic profilings reveal distinct cellular responses during Tilapinevirus tilapiae entry and replication"

_PeerJ, doi:10.7717/peerj.18923_

## Round 0.1 · original submission · Minor Revisions

The study addresses important gaps in our understanding for TiLV-host interactions in fish proteomics. The manuscript may be accepted following minor revision.

·

Basic reporting

The authors have provided sufficient background and references to justify the work conducted. The article is really well structured with an overall good flow of ideas and a very clear and simple language understandable to most readers in the field. The quality of the English language is very good.
Only very few remarks will need attention, such as:
• Line 232: I think you mean “TiLV RNA was detected at 10 mpi”, not 10 hpi. Because in Figure 1b, you don’t have 10hpi.
• Line 237: throughout not “thoughout”

Experimental design

The experimental design section is well organized and detailed. Only minor aspects will require some attention:
• For the immunofluorescence assay, did you include a control consisting of cells stained with the secondary IgG antibody only? Given that your primary antibody is a polyclonal antibody, it would be best to also have a control probed with rabbit pre-immune serum.
• Lines 97 - 98: “ 50µl of lysates ” how was this lysate prepared? I think it is worth mentioning it.
• E-11 cells are known to be constitutively (persistently) infected by a C-type retrovirus called the snakehead retrovirus (SnRV). How did you control for the background impact of this SnRV infection on the true proteomic and phosphoproteomic profile resulting from infection with TiLV?
Because it might be possible that this persistent “preinfection” with SnRV has an impact on the proteomic and phosphoproteomic landscape of E-11 cells in relation to host cell-TiLV interactions.

Validity of the findings

In the current manuscript, the authors really took good care of limiting their conclusions to the supporting results. I only have very few comments on the validity of their findings, which are, according to me, very relevant for the field.
• Line 260 – 261: You stated that “PLS-DA analysis of the phosphoproteome data did not reveal distinct clusters among the E-11 cells collected at different time points”. And indeed in Fig.3d there is no clear distinction between the various time points. I am not familiar with PLS-DA analysis, but have you performed Principal Component Analysis (PCA) with these datasets? This could help to first evaluate the suitability of the biological replicates and then to maybe identify and delineate the factors (primary and secondary) that could contribute to the most substantial differences?
This comment is also valid for the PLS-DA analysis of the phosphoproteomic profile of RHTiB at different time points (Fig. 5d).
• Lines 391-392: “at 10 mpi, there was a suppression of the protein PPP2R2A, an enzyme involved in the dephosphorylation of intracellular proteins”. This finding is very relevant in the sense that if TiLV suppresses PPP2R2A involved in dephosphorylation, it means the virus probably wants to maintain cellular proteins phosphorylation. Which could imply that phosphorylation may be playing an important role during viral infection, especially at early stages of infection.
• Lines 400 – 401: “at 10 mpi, the TiLV-infected RHTiB cells showed inhibition of cytoskeletal proteins”.
It has been shown that the efficient entry of TiLV in TmB cells is dependent at ~80% on actin polymerization and microtubules (Abu Rass R, Kembou-Ringert JE, Zamostiano R, Eldar A, Ehrlich M, Bacharach E. Mapping of Tilapia Lake Virus entry pathways with inhibitors reveals dependence on dynamin activity and cholesterol but not endosomal acidification. Front Cell Dev Biol. 2022 Dec 16;10:1075364. doi: 10.3389/fcell.2022.1075364.). How do you reconcile this previous finding with the Inhibition of cytoskeletal proteins observed in this current study especially at an early entry stage?
• Lines 441 – 442: “The consistency of data across different cell lines and the identification of similar regulatory mechanisms and functions further supports our findings” I don’t think that this is accurate.
In E-11 at 10 mpi, you found 24 proteins and 33 phosphoproteins significantly suppressed, which largely surpasses the 4 proteins and 1 phosphoprotein which were significantly suppressed in RHTiB cells at the same time point.
Similarly in E-11 at 24 hpi, you found 1 protein and 51 phosphoproteins upregulated, as opposed to 4 proteins and 2 phosphoproteins upregulated in RHTiB cells at the same time point. So there is a great difference in the number of modulated proteins as well as in their functions.

Additional comments

It is timely that studies, aiming at elucidating the host cellular responses during TiLV infection, should be undertaken. So the work here presented is a very good start to pave the way to further host cell-virus interactions at the proteomic level. It is therefore very novel, and it advances the field of TiLV research.
It is true that the authors acknowledged the limitations of their work and gave ample justifications regarding these limitations, I however felt a bit unfulfilled in the sense that data analyses could have gone deeper to identify and state which specific main cellular proteins change in phosphorylation status due to the infection (become phosphorylated) and how is the evolution of the virus-induced phosphorylation over time.
This is important because protein phosphorylation could imply protein activation. Such cellular factors undergoing significant phosphorylation during viral entry (at early stages of infection in general) could have a significant impact on viral infection efficiency, meaning that they could have either proviral or antiviral activity. They could thus be potential host-directed targets for the development of antiviral therapeutics.
It is a very good amount of data generated that could be exploited if the authors used the right analyses tools such as kinase-substrate enrichment analysis (KSEA) for example. But I also understand that protein and phosphosprotein databases specific for fish proteomic data analyses and annotations are still crucially lacking, which may significantly hamper this field of research.

·

Basic reporting

The study offers a comprehensive analysis of how Tilapia Lake Virus (TiLV) manipulates the host cellular pathway through the utilization of proteomic and phosphoproteomic techniques. They have gather a sufficient background, literature survey and references. Their finding validate their questions.

Experimental design

Experimental design seems to be efficient. Method and material section needs some improvement and detailed description.

Validity of the findings

The data has provided the well design study outcome. Some places lacking the details where they should provide the number of n (n=?) and how many time experiments has replicated. Some places missing the statistical details description. Supplementary data has provided to support the study.

Additional comments

The authors of this study have confirmed the finding that Tilapia lake virus (TiLV) employs various strategies to acclimate to the host environment, depending on the type of cell. They have tried to explore the molecular mechanism underlaying the host-virus interactions by using proteomic and phosphoproteiomic approach. They utilized the E11 and RHTiB cell lines for their research, uncovering that TiLV infection in E-11 cells involves the inhibition of the FADD-TRAF and JAK-STAT pathways after 24 hours post-infection, whereas in RHTiB cells, it suppresses the AMPK pathway.

Here are some suggestions to improve the quality of manuscript.

Please provide the more details of the importance of choosing E-11 and RHTiB piscine cell lines over other cell lines for this work in introduction.

Please improve the line 65 -line 67 in introduction with more references where phosphoproteomic analysis studies has provided the better approach for identifying the novel mechanisms.

Please provide the catalog no for all reagents used in this study. Method section needs more descriptive incorporation of material and applied method to replicate the things.

Line 131-146 – What is the control medium used for IFC?

Please provide the details for the ANOVA. What ANOVA approach has performed for this study with detailed analysis parameters.

Please give the reference for which tool was used for the heatmap visualization.

Please improve the image quality of Fig 1a and enlarge the section of capture image to better present the cell phenotype in control and 24hpi sample for both E-11 and RHTiB cell lines.

Fig 1c- Please improve the quality of image. Both individual and overlap images should be used here for fig 1C, enlarge one area, and point out arrow for positive TiLV signal by just showing positive signal.

Fig 2b- Please provide E-11 and RHTiB overlapping phosphoproteins as a supplementary table and discuss this in result section.

For Protein Protein interaction whether the differentially expressed genes were used? Please provide ethe details in line 218-line 221. What confidence and algorithm were used for the analysis to better adapt the method.

Please improve the heatmap figure by separating the groups with lines. As the horizontal axis are unreadable its better to omit the horizontal axis and only having groups name on horizontal axis will be better. What method was used for heatmap construction such as unsupervised, cluster or Euclidean method. Please provide all details in method section.

Reviewer 3 ·

Basic reporting

The study design is well explained and the results are supported by the study design. Nevertheless, it is recommended that the authors attend to a few comments outlined below.

Experimental design

Materials & Methods
Lines 88-90, add passage number of E-11 cells used for the study.
Line 99, write concentration of sodium dodecyl sulfate used.
Line 100, which protease inhibitors used, for example eukaryotic? And concentration?
Lines 125: Are these primers designed by you? Add PCR product size. What was the efficiency of primers?
Line 126, did you use neat 4 µl of cDNA template or diluted?
Line 200, add accession number of reference proteome database for the studied fish species.
Data processing and analysis: did you use cRAP (common Repository of Adventitious Proteins) during data analysis? Additionally, authors did not provide any information on database searches such as name of whole genome and accession number along with taxonomy id and entries.

Validity of the findings

Discussion: Please pay more attention to improve the quality of discussion.
Conclusions: Write solid conclusions of the present findings.

Additional comments

Figure 1A: Please provide high magnification of cell culture pictures to see the clear visibility of virus effect on the cells at least 20X or 40X.
Figure 1B: Increase the font size thickness of lines, letters and numbers.
Figures 2, 3 and 5: Please change the colour combination and increase the font size circles, letters and numbers.

·

Basic reporting

The authors provide consecutive sets of experiments concerning proteomic and phosphoproteomic analyses to differentiate between the cellular response of E-11 and RHTiB cells to TiLV infection and the possibility of application for controlling virus infection through improving antiviral drugs or enhancing immune response to facilitate TiLV control. Although the manuscript is interesting and significant minor revisions are needed before publication.

Your introduction needs more detail. I suggest that you add paragraph(s) about the replication process, receptors, and distribution of TiLV in different organs. Genome structure of TiLV. Role of proteomics and phosphoproteins in fish immune responses and viral infection.

Experimental design

Material and methods
The authors provided a detailed methodology for all experiments that were carried out however some points need to be revised:
1- Line 90: ….. (n=3) should be (n=2) since only two cell lines were used
2- Line 137: why authors excluded 6h infection?
3- Line139: (2% ……… please remove brackets
4- Line 140: (PBS)) remove extra brackets

Validity of the findings

Results:
The authors provided results for all experiments with clear and good-quality figures and detailed supplementary data mading the work clear and understandable. Although minor points need to be revised:
1- Fig 2a name of RHTiB must be justified as one line word. E11 …. Please change the color and position to recognize the word easy
2- Fig 2 legend d) Rib ….. please correct to RHTiB
Discussion
1- Line 315: “Viruses are microorganisms that require host cells for their replication” correct to “Viruses require host cells for their replication”
2- Line 317: “…………… regulation to be understood and provides insights” correct to “…………. regulation to be understood and provide insights
3- Line 337: ” indicating that the virus can enter fish cells rapidly” change to ” indicated that the virus can enter fish cells rapidly”
4- Line 346: ” Mahamad et al., 2024” change to “Mohamad et al., 2024”

Additional comments

References
Please follow the journal recommended style.

---

## Round 0.2 · accepted · Accept

Current version addresses all of reviewers comments

·

Basic reporting

The authors have nicely incorporated all the suggestions I have made, and they have improved the quality of the language. So I have no further suggestions or comments regarding this aspect.

Experimental design

My comments on the experimental design have also been nicely addressed. I have no further comments.

Validity of the findings

Here also the authors nicely addressed my suggestions. The suggested PCA analysis was conducted and the results align with the findings of their PLS-DA analysis.

Additional comments

The authors have nicely addressed all the comments and suggestions I previously made. They have also given ample justification to the choice of PLS-DA analysis over PCA analysis. I am overall satisfied by the revisions made.

Reviewer 3 ·

Basic reporting

The authors have made all the necessary edits. The manuscript now appears much better readable.

Experimental design

no comment

Validity of the findings

no comment

Additional comments

no comment